# Effects of Dietary Supplementation of Essential Oils, Lysozyme, and Vitamins’ Blend on Layer Hen Performance, Viral Vaccinal Response, and Egg Quality Characteristics

**DOI:** 10.3390/vaccines12020147

**Published:** 2024-01-30

**Authors:** Tilemachos Mantzios, Ioanna Stylianaki, Soumela Savvidou, Stella Dokou, Georgios Α. Papadopoulos, Ioannis Panitsidis, Apostolos Patsias, Jog Raj, Marko Vasiljević, Marko Pajić, Luis-Miguel Gomez-Osorio, Jasna Bošnjak-Neumüller, Vasilios Tsiouris, Ilias Giannenas

**Affiliations:** 1Unit of Avian Medicine, Clinic of Farm Animals, Aristotle University of Thessaloniki, Stavrou Voutyra 11, 54627 Thessaloniki, Greece; mantziost@yahoo.com (T.M.); biltsiou@vet.auth.gr (V.T.); 2Laboratory of Pathology, Faculty of Veterinary Medicine, Aristotle University, 54124 Thessaloniki, Greece; stylioan@vet.auth.gr; 3Research Institute of Animal Science, Hellenic Agricultural Organisation-Demeter, 58100 Giannitsa, Greece; savidousumela@gmail.com; 4Laboratory of Nutrition, Faculty of Veterinary Medicine, Aristotle University, 54124 Thessaloniki, Greece; dokoustella@vet.auth.gr (S.D.); ipanits@vet.auth.gr (I.P.); 5Laboratory of Animal Husbandry, Faculty of Veterinary Medicine, Aristotle University, 54124 Thessaloniki, Greece; geopaps@vet.auth.gr; 6Agricultural Poultry Cooperation of Ioannina “PINDOS”, Rodotopi, 45500 Ioannina, Greece; apatsias@pindos-apsi.gr; 7PATENT CO. DOO, 24211 Misicevo, Serbia; jog.raj@patent-co.com (J.R.); marko.vasiljevic@patent-co.com (M.V.); luis.gomez.osorio@alura.bio (L.-M.G.-O.); jasna.bosnjak@patent-co.com (J.B.-N.); 8Department for Epizootiology, Clinical Diagnostic, Pathology and DDD, Scientific Veterinary Institute “Novi Sad”, Rumenački Put 20, 21000 Novi Sad, Serbia; markopajic@niv.ns.ac.rs

**Keywords:** laying hens, oral vaccination, plant extracts, lysozyme, egg quality, antioxidant activity, trachea evaluation

## Abstract

Maintaining respiratory tract health is crucial for layers, impacting gut health, laying performance, and egg quality. Viral diseases and standard vaccinations can compromise tracheal epithelium function, leading to oxidative stress. This study assessed the impact of a blend of feed additives, predominantly lysozyme (L), essential oils (EO), and vitamins (VIT) (referred to as L + EO + VIT), on young layers during an oral vaccination schedule. The supplementation significantly enhanced antibody titers for Newcastle Disease Virus (NDV) and Infectious Bronchitis Virus (IBV) after vaccination, trachea functionality and intestinal health in the jejunum, increased egg production, and exhibited a trend toward higher egg weight. Although feed intake showed no significant difference, egg quality remained consistent across experimental groups. Moreover, L + EO + VIT supplementation elevated total phenolic content in eggs, improving oxidative stability in both fresh and stored eggs, particularly under iron-induced oxidation. Notably, it substantially reduced yolk lipid peroxidation and albumen protein carbonyls. In conclusion, water supplementation with L + EO + VIT may enhance humoral immune response to IBV and NDV, positively impacting hen productivity. These findings indicate improved tracheal function and enhanced oxidative stability, emphasizing the potential of this blend in promoting overall health and performance in layers.

## 1. Introduction

Respiratory diseases control is crucial in all types of poultry including laying hens. Respiratory infections in layers can considerably impair bird health and welfare, as well as performance and hence profitability of flocks [1]. Although in broilers respiratory diseases mostly affect the respiratory tract itself, in layers respiratory infections affect both the airways and the oviduct [2]. Therefore, respiratory disease can have a significant economic impact on layer production due to increased mortality, decreased egg production, shell quality issues, and increased costs of production associated with diagnostic and treatment procedures [3]. To mitigate the impact of respiratory diseases on egg production, it is crucial to implement proper preventive measures. This relies on a combination of effective biosecurity and good farming management supporting bird health and immune system integrity [4]. Vaccination against respiratory pathogens is also an effective strategy in reducing the incidence and severity of respiratory diseases, thereby safeguarding egg production. However, several live viral vaccines may affect the trachea functioning, egg production, and/or humoral response [5].

The use of prophylactic antibiotics in laying hens has been applied to decrease bird morbidity and mortality as well as to improve productivity of the flock. However, some drawbacks of antibiotics have been documented, including antimicrobial drug resistance that impacts the human population [6,7]. Natural alternatives such as phytobiotics, which comprise essential oils, plant extracts, and medicinal herbs, exhibit a wide range of biological activities [8,9]. EO use either as feed or water supplements has shown a variety of desirable effects [10]. Either prophylactic or therapeutic effects of EO in laying hens improve performance, immunological outcomes, and eggshell quality [11]. Additionally, Olgun (2016) showed that the addition of an EO mixture had a positive impact on egg weight, egg mass, and eggshell thickness [12]. EO exhibits a beneficial influence on lipid metabolism. Research on the incorporation of essential oils into laying hens’ diets has revealed that the inclusion of polyunsaturated fatty acids in the egg yolk tends to increase, whereas the levels of saturated and monounsaturated fatty acids eventually decrease [13].

Natural monoterpenes, such as p-cymene and γ-terpinene, abundantly present in essential oils like thyme and oregano, serve as precursors to thymol [14]. Extensive research indicates that these terpenes exhibit antimicrobial and antioxidant properties, potentially enhancing gut health in chickens and contributing to the overall well-being and performance of chickens [15,16]. Likewise, linalool, another monoterpene which can be found in lavender and coriander essential oils, has been explored for its potential to modulate stress responses and enhance the immune system in chickens [17]. Lastly, carvacrol and thymol are potent bioactive compounds, renowned for their antimicrobial, antioxidant, and anti-inflammatory properties [18]. Their dietary inclusion in poultry nutrition has been associated with improved gut health, reduced pathogenic load, and enhanced overall performance [19,20].

Nowadays, the use of lysozyme as a growth and health enhancer has been a milestone in animal nutrition. Lysozyme is a naturally occurring antimicrobial peptide (AMP) belonging to the innate immune system [21]. By hydrolyzing the 1,4-glycosidic bond between N-acetylmuramic acid and N-acetyl glucosamine of the bacterial cell wall, it shows bacteriolytic activity, mostly against numerous Gram-positive bacteria [22]. According to a report, lysozyme could reduce the intestinal lesions caused by *C. perfringens* [23]. Moreover, several papers are providing evidence of immunomodulatory effects of lysozyme and its positive impact on specific humoral immunity [24,25]. A combination of lysozyme with EO compounds may offer a natural promising alternative to replace antibiotics and hence to enhance humoral immune response in the respiratory tract health.

Newcastle disease is a highly contagious viral disease affecting several species of birds and enormous efforts have been made at controlling and understanding its epidemiology and virology. Newcastle disease is ranked as the fourth most significant poultry disease according to the World Livestock Disease Atlas after examining more than one hundred countries. Newcastle disease has a multi-system impact, leading to a mortality rate of up to 100% in unvaccinated birds. The economic consequences are substantial, attributable to trade limitations and embargoes imposed on regions or nations affected by the disease. It occurs worldwide, while some strains are more prevalent in certain areas of the world than others. The control measures, which are imposed by legislation, involve the implementation of stamping out policies and restriction zones causing severe consequences for trade [26]. Another significant financial impact is attributed to the avian coronavirus causing infectious bronchitis (IBV), a disease that harms the upper respiratory tract of chickens. It is an economically important, highly contagious, acute, upper-respiratory tract disease of chickens, caused by the avian coronavirus infectious bronchitis virus. The virus is globally distributed and spreads through inhalation or direct contact with infected birds, as well as contaminated litter, equipment, or other objects. Depending on the viral strain, infections can result in acute upper-respiratory tract diseases, declines in egg production, compromised egg quality, and nephritis [27].

This study aimed to determine the impact of a water-soluble product combining the benefits of EO, lysozyme (L), and vitamins (VIT) in young laying hens’ performance and egg quality outcomes, as an alternative blend to pharmaceutical feed additives. Furthermore, we evaluated the affect of this alternative blend of L + EO + VIT on trachea epithelium integrity and humoral response following oral vaccination in a commercial facility.

## 2. Materials and Methods

### 2.1. Determination of Active Ingredients of L + EO + VIT Blend

The quality control of the main compounds of the product was performed using Repeatability (within-laboratory) RSDr (%) (Table 1) and Reproducibility (within-laboratory) RSD_WLR_ = CV (%) methodologies (Table 2). A total of 12 samples of the product were mixed gently. A portion of 0.2 g of homogenized sample with an accuracy of 0.001 g was put into six 15 mL conical tubes. For the extraction, 10 mL ethanol (Panreac Applichem, Barcelona, Spain) was added to the vessel. The mixture was shaken in an orbital shaker at 200 rpm for 15 min at room temperature. After extraction, a dilution process was conducted with hexane (Panreac Applichem, Barcelona, Spain) using for the first dilution 100 µL sample + 900 µL hexane and for the second 200 µL sample (D1) + 800 µL hexane.

### 2.2. Animals, Diets, and Experimental Design

Authorization of the experimental protocol was applied by the Research Committee of Aristotle University, Thessaloniki, Greece, under the authorization code 73380/2023. Throughout the trial, the birds were handled in compliance with local laws and regulations (PD, 2013). One hundred and sixty Lohmann Brown-Classic laying hens (24-week-old) from a flock of 8000 hens kept at a commercial poultry farm (Kotopoula Barbagianni), in Axos, Giannitsa, Pella (latitude 40.77°, longitude 22.45°) were used in this study. Birds were allocated into four treatment groups, from weeks 24 to 32, consisting of four replicates per group with furnished pens (length 3.6 m; width 0.4 m; height 0.3 m) housing 10 hens each, placed in small battery cages equipped with bell-type drinkers. The experimental design was as follows: Group CTR: to which birds were not vaccinated and received drinking without the L + EO + VIT blend; Group CI: to which birds were unvaccinated and received drinking water supplemented by the L + EO + VIT blend at the level of 1.0 g/L; Group CV: to which birds were vaccinated and received drinking without the L + EO + VIT blend; Group CVI to which birds were vaccinated and received drinking water supplemented by the L + EO + VIT blend at the level of 1.0 g/L. Diet composition is presented in Table 3. Vaccination against infectious bronchitis (IB) and Newcastle disease (ND) was applied at week 25. The hens of the CV and CVI groups were immunized with the commercial Nobilis^®^ MA5+ Clone 30 (MSD, Animal Health, Haarlem, The Netherlands) live attenuated strains. The selected vaccine and the tested product were administrated through the bell-type drinkers. During the experimental phase, unrestricted access to feed and water was maintained. The lighting duration was consistently set at 16 h of continuous light per day. Temperature control ensured a range of 17 °C to 23 °C, while relative humidity was fine-tuned within the range of 65% to 75%. Egg samples were gathered on days 14, 28, and 42 of the study. Following the conclusion of the experimental period, all collected eggs were stored at 4 °C, awaiting subsequent processing.

### 2.3. Egg Yield and Egg Quality Parameters

Daily records were maintained for hen performance, egg production, and feed intake. Evaluations of egg quality parameters took place in the final two days of the trial, specifically during the 32nd week of age, evaluating eight eggs per replicate. Eggshell breaking strength (N) was assessed by an Instron 5542 K Model (Bucks, UK) as compression strength of fresh eggs. Following the egg cracking process, the weights of the egg albumen, yolk, and shell were documented, along with measurements for yolk height and egg diameter. The precision thickness caliper, with a precision of 0.01 mm, was utilized to measure eggshell thickness after removing shell membranes at the equator. Egg yolk and eggshell color were assessed using a Miniscan XE chromameter (HunterLab, Reston, VA, USA), configured on the L* (lightness), a* (redness), and b* (yellowness) system. Calibration was performed using white and black tiles.

### 2.4. Determination of Egg Yolk Oxidative Stability in Fresh and Stored Intact Eggs, Total Antioxidant Capacity, as well as Protein Carbonyls Formation in Albumen and Yolk with or without Iron Induced Challenge

The stability of egg yolk against oxidative processes was appraised using the malondialdehyde (MDA) measurement method detailed by Ahn et al. [28], with slight adaptations tailored to the specific parameters of this study.

In order to magnify the oxidative phenomena, egg yolk and albumen were separated and each one was mixed with a solution containing iron and ascorbic acid, to provoke oxidation as described by Kornsbrust and Mavis [29] with slight modifications. This method demands the incubation of the mixture of egg yolk or albumen with the oxidant solution at 37 °C for 30 min. The oxidant solution is made by adding 1.138 mM ferrous sulphate and 0.368 mM ascorbic acid per 1.0 mL of water. Following the incubation period, both the iron-induced subsamples and the non-induced subsample underwent an immediate malondialdehyde assay to quantify the degree of lipid oxidation.

To assess protein carbonyls, we utilized the procedure outlined by Patsoukis et al. [30], both on unchallenged egg yolks without oxidation and on egg yolk samples subjected to oxidation challenges with iron and ascorbic acid.

The same samples were subjected to the determination of total antioxidant capacity using the method outlined by Prietto et al. [31], employing a phosphomolybdate reagent.

Lastly, the total phenolic content of each chicken egg yolk sample was determined using the assay reported by Shang et al. [32]. A total of 2 g of egg yolk powder with or without iron-induced oxidation were homogenized with the use of a vortex in 8 mL of MeOH (Panreac Applichem, Barcelona, Spain). Following that the sample was centrifuged and the supernatant was collected, quantified, and combined with 400 μL of 20% TCA (VWR Chemicals, Leuven, Belgium) to make the diluted sample and again centrifuged for 20 min. In total, 125 μL of the sample was combined with 125 μL of Folin-Ciocalteu (Panreac Applichem, Barcelona, Spain) reagent and incubated at room temperature for 30 min. Following the incubation phase, the determination of total phenolic content was conducted by utilizing a spectrophotometer (UV-1700, Shimadzu, Japan) set to measure absorbance at 725 nm.

### 2.5. Monitoring Serum IBV and NDV Antibody Titers

Blood samples were collected from 16 layers per group and the serum was submitted for IBV and NDV antibodies quantification using Enzyme-Linked Immunosorbent Assay (ELISA) commercial kits for IBV (CK119 IBV, BioChek Limited, Surrey, UK) and NDV (CK116 ND, BioChek Limited, Surrey, UK), respectively. Humoral response was measured using antibody titers determination at the beginning of the trial. Then, a new vaccination for Infectious Bronchitis Virus (IBV) and Newcastle Disease Virus (NDV) was performed during the first week (25th week of age) of the trial. Antibody titers were then quantified in serum samples on days 14 and 28 after vaccination (27th and 29th week of age).

### 2.6. Blood Parameters and Serum Antioxidant Profile Evaluation

Blood sampling was conducted at the 29th week of age, with eight hens randomly chosen from each group. The samples were collected through the brachial vein into tubes containing both EDTA and without anticoagulant (BD Vacutainer^®^, Plymouth, UK) to obtain serum. Total white blood cells (TWBC), heterophils, lymphocytes, monocytes, eosinophils, glucose, albumin, cholesterol, and total protein were analyzed using an automatic blood biochemistry analyzer. The samples collected without anticoagulant were centrifuged at 3000 RPM for 15 min and serum was stored at −20 °C for antioxidant indices evaluation. Antioxidant enzyme production was assessed by measuring Thiobarbituric Acid-Reactive Substances (TBARS), Catalase (CAT), Superoxide Dismutase (SOD), and Glutathione Peroxidase (GPx) using commercial ELISA kits (Siemens Healthcare GmbH, Erlangen, Germany).

### 2.7. Evaluation of Liver Histopathology and Small Intestine Histomorphometry

Eight hens per group were randomly selected at the end of the experimental period for histomorphological evaluation. The birds were euthanized by exposure to a rising concentration of carbon dioxide in an air-tight container and were subjected to necropsy. Liver evaluation was based on the present or absence of epithelial hyperplasia and hemorrhagic lesions. The observed histopathological findings were assessed using a semiquantitative scoring system as follows: absent/minimal (score = 0), mild (score = 1), and severe (score = 2). Samples from intestinal segments of the duodenum and jejunum, extracted from each bird and flushed with 0.9% saline, underwent histomorphometric examination. The gut segments were fixed in Carnoy’s solution for morphometric analysis. The tissues were routinely embedded in paraffin wax blocks, sectioned at a thickness of 5 μm, mounted on glass slides, and stained with Haematoxylin and Eosin (HE). The morphometric indices assessed included villi height (Vh), measured from the tip of the villus to the crypt, crypt depth (Cd), measured from the base of the villus to the submucosa, and their respective ratios [33]. Morphometric analyses from duodenum and jejunum were performed on 10 well-oriented intact villi as well as on 10 crypts [34].

### 2.8. Histopathology of the Respiratory Tract (Trachea and Lungs)

Tracheal segments (4–5 cm in size) and lung tissues were sampled from two hens per replicate 7 and 14 days after vaccination (25th and 26th week of age) and fixed in a buffered formalin dilution of 10%. Five tracheal rings per hen were processed, sectioned, and stained with hematoxylin and eosin (H and E) for histopathological determination produced by the vaccine challenge. The modified protocol of Alvarado et al. [35] scoring 1–4 and focusing on epithelial hyperplasia, lymphoplasmacytic infiltration, ciliated cell, and mucous gland morphology was used for this histopathological evaluation.

### 2.9. Statistical Analyses

The data were processed using IBM SPSS Statistics Ver.20 software, designating the replication (pen) as the experimental unit for performance outcomes, encompassing egg and blood analyses. One-Way Analysis of Variance (ANOVA) was employed, with dietary treatments serving as the grouping factor. The results are presented as mean with the pooled standard error of mean (SEM). Statistical significance was determined at a probability level of *p* < 0.05. In cases of significance, means were differentiated using Tukey’s B test.

## 3. Results

### 3.1. Hen Performance and Egg Quality Characteristics

The effects of the dietary supplementation with the L + EO + VIT blend on egg production are shown in Table 4. Egg production did not differ among the groups at the starting line of the experiment (*p* > 0.05). The CI and CV groups showed higher values during week one compared to the CTR and CVI groups (*p* ≤ 0.05), while during week three the CV and CVI groups had significantly increased egg production compared to the other groups (*p* ≤ 0.05). Egg production values did not differ among groups at the final stage of the trial (*p* > 0.05). L + EO + VIT blend supplementation resulted in similar values regarding egg parameters among the different groups (*p* > 0.05) (Table 5). However, the CVI and CTR groups showed a borderline significance for higher eggshell thickness compared to the CI and CV groups (*p* = 0.061).

### 3.2. Egg Oxidation

Table 6 presents the effects of the blend L + EO + VIT supplementation on egg yolk TBARS values. Both groups CI and CVI that were provided with the blend L + EO + VIT showed significant lower yolk TBARS values compared to the unsupplemented groups CTR and CV in fresh and stored yolk samples and also after the performance of the oxidation challenge (*p* < 0.001). Total antioxidant capacity in fresh yolk is shown in Table 7. TAC was higher in both fresh yolk and fresh yolk after the oxidation challenge of CI and CVI compared to the other two groups (*p* < 0.001). The results of yolk total phenolic content are presented in Table 8. Group CI and CVI that were orally supplemented with the blend L + EO + VIT showed the highest TP content (*p* < 0.001) in comparison with CTR and CV. Finally, yolk protein carbonylation was significantly higher for the unsupplemented groups CTR and CV versus the rest of the treatments (*p* < 0.001) (Table 9).

### 3.3. Antibody Titers for Newcaste Disease and Infectious Bronhitis

The antibodies titers production against NDV and IBV are shown in Table 10. Before vaccination, all birds had similar antibody titers regarding both diseases (*p* > 0.05). However, at the interval of two-weeks post-vaccination, groups CV and CVI which were vaccinated had higher antibody titers for NDV and IBV compared to the unvaccinated groups CTR and CI (*p* < 0.001). Regarding week four post-vaccination, both vaccinated groups (CV and CVI) showed higher serum antibodies for IBV (*p* < 0.001). On the contrary, the levels of antibody titers for NDV were significantly higher for the CVI group that was orally supplemented with both the vaccine and the natural product compared to all three groups (*p* < 0.001).

### 3.4. White Blood Cell Count and Antioxidant Capacity

White blood cell counts and serum outcomes are presented in Table 11. Total white blood cells (TWBC) were significantly increased for the CVI group compared to the CV group (*p* < 0.05). Glucose concentrations were higher for the CTR and CV group, while the CI group had the lowest values (*p* ≤ 0.05). Additionally, all groups showed similar values for the other hematological and serum parameters (*p* > 0.05). Oxidation status indices are presented in Table 12. The serum of both vaccinated groups CI and CVI showed significantly lower MDA values compared to the CV group (*p* ≤ 0.05). Regarding Glutathione Peroxidase (GPx), the vaccinated birds supplemented with the L + EO + VIT blend (CVI) presented the lowest values among the groups and significantly lower than the CTR group (*p* ≤ 0.05). SOD values did not differ among the experimental groups (*p* > 0.05), while there was a borderline significance for groups CI and CVI to have lower catalase values compare to the CTR and CV groups (*p* = 0.058).

### 3.5. Liver and Lung Histopathology and Small Intestine Histomorphometry

Dietary supplementation effects on liver and lung lesions are displayed in Table 13. No differences were found among groups (*p* > 0.05). L + EO + VIT blend supplementation effects on small intestine histomorphometry are presented in Table 14. Similar values were evident regarding duodenum villus height and crypt depth, as well as jejunum villus height (*p* > 0.05). In contrast, the CVI group had significantly lower jejunum crypt depth compared to the control group (CTR) (*p* = 0.034).

### 3.6. Trachea Histological Lession Score

The impact of vaccination, the supplementation of drinking water with the L + EO + VIT blend, and their combination on the trachea histological lesion score are presented in Table 15 and Figure 1. Regarding the first trachea evaluation, one week after the performance of vaccination, the CVI group had significantly lower lesions compared to the CTR and CI groups (*p* = 0.009). Similarly, the hens of the CVI group had the lowest lesions two weeks after vaccination compared to the CTR group (*p* = 0.009).

## 4. Discussion

In the present experimental trial, the potential beneficial effects of a synergetic product—composed of EO, lysozyme, and vitamins—on overall hen efficiency and health were evaluated. Driven by the urge to search for natural alternatives to replace antibiotics in laying hens concurrently with performance maximization, natural compounds are steadily rising; however, a variety of concerns arise in the available literature [13,36]. The current genetic lines of laying hens have been proven to be quite resistant to metabolic problems, such as susceptibility to osteoporosis or liver failures, when proper management and nutrition are provided [37]. Furthermore, respiratory issues may alter layers’ homeostasis thus compromising laying performance [38]. To our knowledge, no literature data exist on the effects of phytogenic compounds including lysozyme or vitamins on supporting vaccination against NDV or IBV in laying hens. In broilers, one study conducted a decade ago suggesting supporting broiler chickens by using the EO of eucalyptus and peppermint for boosting humoral response through vaccination against velogenic NDV strain, with this approach underlying their survivability and performance [39]. Commercial hens infected with NDV or IBV can exhibit both respiratory and reproductive system issues and economic losses because of reduced egg production, lower egg quality, moderate to high mortality, and deterioration of health and welfare. Although the gold standard approach for application of the NDV or IBV vaccines is oculonasal [40], this route is impractical for mass administration; thereby, live vaccines are typically administered via drinking water [41].

According to our results, the performance of IBV and NDV vaccination οn a flock of young laying hens had a positive effect on egg production three weeks after administration; although for a short timeframe vaccination stress led to a minor decrease in the production rate. A positive effect of phytobiotics has also been reported on egg production [42,43]. While a few studies have reported positive effects of certain EOs on egg production, others have found no significant impact [36]. However, the difference in egg production observed at week one could be attributed to the relocation from enriched cages to land-based cages to vaccinate groups of hens separately. Various factors such as the type of EO, dosage, method of administration, and the specific conditions of the study can influence the results. The administration of the natural product did not influence egg laying yield, although overall birds’ performance was very sufficient. Egg quality characteristics remained similar among the different groups even though there was a tendency for increased eggshell thickness for the supplemented plus vaccinated group. The EO thymol was reported to improve mineral absorption thus amplifying eggshell thickness; nonetheless, the mode of action has not been entirely elucidated [13,36].

It was expected that L + EO + VIT blend supplementation would positively affect lipid oxidation status of the eggs. The main secondary metabolites formulating in the tested blend of additives were carvacrol, thymol, p-cymene, γ-terpinene, and linalool, all proven to exhibit antioxidant effects [44,45,46,47,48]. According to our results, both groups provided with the L + EO + VIT blend showed lower levels of lipid oxidation in the produced eggs. Various authors reported that EO’s bioactive molecules function as free radical binders and inhibit lipid peroxidation, thus reducing the risk of oxidative stress and lipid oxidation [49]. Moreover, evidence supports the antioxidant effects exhibited by lysozyme and various vitamins [50,51]. Additionally, water supplementation with the L + EO + VIT blend led to decreased TBARS serum concentration in birds, indicating improved endogenous antioxidant capacity. These results are in accordance with previous reports supporting the effect of phytobiotics as antioxidants [43].

Storage of intact eggs did not have a significant effect on lipid oxidation. This can be explained due to the presence of antioxidant compounds that preserve the flavor and quality characteristics of eggs during an extended storage [52]. The phosvitines appear to effectively protect yolk lipids [53]. It is possible that the protective compounds of the albumin are not the same as the yolk. The eggs of the hens that did not consume the L + EO + VIT blend showed enhanced oxidation as was found by three different oxidation techniques. When an iron solution was added, rapid oxidation on both the yolk and the albumen of the eggs was observed. However, the L + EO + VIT blend proved to enhance total antioxidant capacity, lipid oxidation of the yolk, and formation of protein carbonyls in the albumen. These results are in agreement with higher total phenolic content in eggs found in the treatment receiving the L + EO + VIT blend, thus supporting a potent sparing effect of the inherit antioxidant components in the egg.

The use of vaccinations is a common prophylactic practice against viral and/or bacterial diseases in poultry. Its utilization in chicken production aims to prevent and minimize the incidence of clinical disease at the farm level, as well as to improve performance. However, a variety of factors, including the system used to raise birds, biosecurity, and management practices can affect the ideal application of poultry vaccines. For example, live attenuated vaccines provided in drinking water can be eliminated if water sanitizers are not removed or many infectious pathogens have different serotypes and vaccine antigens do not provide protection against all field strains [54]. Moreover, immunosuppression stress, mycotoxins, and some infectious agents (including infectious bursal disease virus, infectious anemia virus, and Marek’s disease in chickens, and hemorrhagic enteritis in turkeys) can all impair the immune system [55]. Overall, a successful vaccination program should lead to an improvement in the performance of the flock. The morbidity and mortality rates, as well as other performance measurements including feed conversion, egg production, and egg quality, are useful measurable and comparable indicators to assess the general health of a flock [55].

In the present study, the combination of lysozyme, EO, and vitamins as a water soluble blend of additives improved humoral responses by increasing antibody titers against IBV. Lysozyme is an enzyme that plays a key role in enhancing the avian immune system. It is naturally present in various body fluids and tissues, including tears, saliva, mucus, and egg whites. One of the key functions of lysozyme is its antimicrobial activity. It acts by breaking down the cell walls of certain bacteria, inhibiting their growth and replication [56]. By targeting the bacterial cell walls, lysozyme helps prevent the colonization and infection of the chicken’s respiratory tract as well as gastro-intestinal system by harmful bacteria [57]. Furthermore, lysozyme has been found to have immunomodulatory effects. It can help stimulate the production of immune cells, such as macrophages and lymphocytes, which are part of innate and acquired immunity, respectively [56]. Lysozyme also support the activation of various immune pathways, promoting the production of cytokines such as Interleukin-1, -6, and -8 and other immunomodulatory molecules that contribute to build a robust immune response against pathogens [58]. Despite extensive research on lysozymes in broiler chickens, there is little information available about how it affects laying hens [59].

Furthermore, a single administration of live attenuated NDV vaccine is sufficient to rapidly trigger an immune response inducing immunological memory and thereby protection against infectious disease. Much higher doses of the live attenuated vaccines are administered to protect birds from disease [60]. Nevertheless, opting for this is not financially practical due to the high cost of vaccination per bird. To make it economically viable, there is a need for cost-effective vaccination strategies to address the issue, whether for safeguarding against the virus or mitigating virus shedding linked to traditional Newcastle disease vaccines. A crucial aspect influencing the efficacy of vaccination is the tissue tropism of the vaccine. Traditional live attenuated Newcastle Disease Virus (NDV) vaccines, such as the LaSota strains, primarily target the respiratory system, fostering a stronger mucosal immune response in the airways, the probable site of initial virus exposure [61]. Thus, while the vaccines effectively prevent clinical signs and mortality induced by NDV isolates, their incapacity to halt virus shedding post-challenge results in the ongoing presence of the virulent virus in the environment. Therefore, higher antibody titers are needed to avoid virus entrance to the host, thus providing birds with protection against wild challenges. It is necessary that veterinarians are encouraged to examine the trachea tissue and lungs, as well as the alterations of the gastrointestinal tract. The liver and intestine must be also evaluated for histopathology standpoint. In the current study, the addition of the blend L + EO + VIT into the drinking water improved trachea functionality and morphology, as well as liver functionality. Moving to the intestinal morphometry, the dietary supplementation with the L + EO + VIT blend, either alone or in combination with bird vaccination, does not significantly alter the histomorphometry of the duodenum and jejunum in terms of villus height. However, the observed reduction in crypt depth in the jejunum for the CVI group implies a potential synergistic effect between the dietary blend and bird vaccination in improving the health of this intestinal section.

## 5. Conclusions

In conclusion, layers that received a water-soluble blend of additives, L + EO + VIT blend, at the level of 1 g/L, containing lysozyme and EO as the main compounds and vitamins and other ingredients in smaller concentrations, showed a better humoral immune response and secretory defense after oral challenges using vaccinations of NDV and IBV strains, alongside improved functionality of the respiratory tract. The hens showed no deterioration either in laying performance or egg quality. Viral diseases or even standard vaccination may affect trachea epithelium and trigger oxidative stress. However, this blend of additives can support protective activity against oxidation and underlie trachea functionality.

## Figures and Tables

**Figure 1 vaccines-12-00147-f001:**
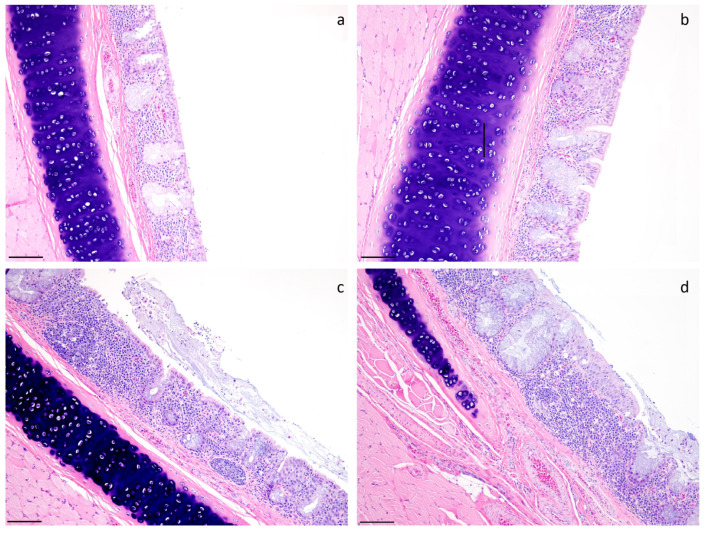
Microscopic changes in trachea taken from the layer hens: (**a**) Score 1—no epithelial lesions and mild inflammatory infiltration, (**b**) Score 2—mild epithelial hyperplasia and inflammatory infiltration, (**c**) Score 3—moderate epithelial hyperplasia and inflammatory infiltration, and (**d**) Score 4—severe epithelial hyperplasia and inflammatory infiltration with cilia loss, according to the scoring system that was used (modified protocol of Alvarado et al. [35]). HE. Bar = 250 μm.

**Table 1 vaccines-12-00147-t001:** Repeatability (within-laboratory) RSD_r_, [%] of the main compounds of the tested product ^1^.

L + EO + VIT Blend	p-Cymene	γ-Terpinene	Linalool	Thymol	Carvacrol
mg/kg	243	113	2330	5202	1841
RSD_r_ (%) Limit	5.3	5.3	3.7	3.7	3.7
Calculated RSD_R_ (%)	3.61	3.29	3.19	2.53	2.50

^1^ the tested product is a blend of additives that dominantly contains lysozyme, essential oils, and vitamins.

**Table 2 vaccines-12-00147-t002:** Reproducibility (within-laboratory) RSD_WLR_ = CV, [%] of the main compounds of the tested product ^1^.

L + EO + VIT Blend	p-Cymene	γ-Terpinene	Linalool	Thymol	Carvacrol
mg/kg	243	113	2330	5202	1841
RSD_R_ (%) Limit	8	8	6	6	6
Calculated RSD_R_ (%)	5.82	5.81	5.49	5.36	4.40

^1^ the tested product is a blend of additives that dominantly contains lysozyme, essential oils, and vitamins.

**Table 3 vaccines-12-00147-t003:** Composition of the control diet.

Ingredients, %	Weeks 24–32
Maize	55.20
Wheat, soft	5.00
Soybean meal	24.0
Wheat bran	3.00
Soybean oil	0.50
Calcium phosphate	1.50
Limestone (Calcium carbonate)	9.50
Salt	0.23
Sodium carbonate	0.22
Lysine	0.13
Methionine	0.32
Threonine	0.05
Valine	0.10
Vitamin and mineral premix ^1^	0.25
Calculated Analysis (As fed basis)	
Metab. Energy, Kcal/kg	2730
Moisture, %	11.61
Protein, %	16.70
Crude fiber, %	3.30
Crude fat, %	2.71
Starch,%	30.07
Ash, %	8.41
Calcium, %	3.65
Total phosphorus, %	0.65
Sodium, %	0.16
Chloride, %	0.16

^1^ Supplying per kg feed: Vitamin A, 10,000 IU; Vitamin D_3_, 2500 IU; Vitamin E, 44.7 IU; 2 mg menadione sodium bisulfite, 2 mg thiamine hydrochloride, 3 mg riboflavin, 4 mg pyridoxine hydrochloride, 0.02 mg cyanocobalamin, 20 mg niacin, 10 mg pantothenic acid, 1.0 mg folic acid, 0.07 mg biotin, 50 mg ascorbic acid, 300 mg choline chloride, and 40 mg carotenoids, 80 mg Zn, 40 mg Mn, 160 mg Fe, 70 mg Cu, 0.25 mg Co, 1 mg I, and 0.2 mg Se.

**Table 4 vaccines-12-00147-t004:** Effect of dietary supplementation with the L + EO + VIT blend on laying hen egg production.

Egg Production %	Groups ^1^				SEM ^2^	*p*
	CTR	CI	CV	CVI		
Week 0	88.85	87.96	88.55	87.82	0.254	0.555
Week 1	83.99 ^b^	88.86 ^a^	89.96 ^a^	83.75 ^b^	0.777	0.002
Week 3	83.24 ^b^	81.71 ^b^	92.97 ^a^	92.93 ^a^	1.104	<0.001
Week 5	89.14	90.01	91.90	90.15	0.522	0.310

^1^ CTR: not vaccinated and received drinking water without L + EO + VIT blend; CI: not vaccinated and received drinking water with L + EO + VIT blend; CV: vaccinated and received drinking water without L + EO + VIT blend; CVI: vaccinated and received drinking water supplemented by L + EO + VIT blend, (n = 4). ^2^ SEM: Standard Error of Mean. ^ab^ Different letters denote significant (*p* ≤ 0.05) differences between the dietary treatments.

**Table 5 vaccines-12-00147-t005:** Effect of dietary supplementation with the L + EO + VIT blend on egg quality parameters.

Egg Characteristic	Groups ^1^				SEM ^2^	*p*
	CTR	CI	CV	CVI		
Egg weight (g)	60.71	60.82	60.77	61.86	0.935	0.971
Yolk weight (g)	16.95	16.07	16.20	16.12	0.193	0.346
Albumen weight (g)	38.18	39.30	39.06	40.15	0.997	0.929
Egg shell weight (g)	5.57	5.45	5.51	5.58	0.103	0.967
Egg shell thickness (mm)	0.479 ^x^	0.428 ^y^	0.441 ^y^	0.478 ^x^	0.020	0.061
Egg lateral index (cm)	59.07	58.97	59.11	59.27	0.436	0.997
Egg diameter (cm)	43.40	43.22	43.27	43.36	0.229	0.994
Egg shape index	73.59	73.34	73.27	73.28	1.377	0.997
Egg shell color	33.02	29.07	32.15	31.88	0.820	0.367
Yolk color	14.12	13.87	13.62	14.12	0.161	0.669
Haugh units	74.57	76.62	74.00	77.87	1.202	0.658
Egg weight in water (g)	4.42	4.54	4.28	4.35	0.327	0.896
Egg weight, specific (g/cm^3^)	1.079	1.081	1.076	1.076	0.002	0.871
**Egg strength**						
Egg shell deformation (N/m^2^)	0.027	0.025	0.024	0.027	0.009	0.610
Egg shell breaking force (N/m^2^)	3719.75	3719.00	3709.50	3733.25	0.533	1.000

^1^ CTR: not vaccinated and received drinking water without L + EO + VIT blend; CI: not vaccinated and received drinking water with L + EO + VIT blend; CV: vaccinated and received drinking water without L + EO + VIT blend; CVI: vaccinated and received drinking water supplemented by L + EO + VIT blend, (n = 4). ^2^ SEM: Standard Error of Mean. ^x,y^ Values in the same row with different superscripts tend to differ (0.05 < *p* ≤ 0.10).

**Table 6 vaccines-12-00147-t006:** Effect of dietary supplementation with the L + EO + VIT blend on oxidative status of egg yolk of fresh or stored intact eggs for 30 days and after challenge with iron and ascorbic acid for 30 min to induce oxidation.

Egg Yolk TBARS	Groups ^1^				SEM ^2^	*p*
	CTR	CI	CV	CVI		
Fresh yolk (ng/mL)	33.5 ^a^	20.2 ^b^	31.5 ^a^	20.8 ^b^	1.086	<0.001
Stored yolk (ng/mL)	43.6 ^a^	28.3 ^b^	44.4 ^a^	22.1 ^b^	1.733	<0.001
Fresh yolk after oxidation challenge (μgGAE/g)	132.2 ^a^	75.5 ^b^	128.8 ^a^	84.7 ^b^	4.588	<0.001

^1^ CTR: not vaccinated and received drinking water without L + EO + VIT blend; CI: not vaccinated and received drinking water with L + EO + VIT blend; CV: vaccinated and received drinking water without L + EO + VIT blend; CVI: vaccinated and received drinking water supplemented by L + EO + VIT blend, (n = 4). ^2^ SEM: Standard Error of Mean. ^ab^ Different letters denote significant (*p* ≤ 0.05) differences between the dietary treatments.

**Table 7 vaccines-12-00147-t007:** Effect of dietary supplementation with the L + EO + VIT blend on total antioxidant capacity after challenge with iron and ascorbic acid for 30 min to induce oxidation on egg yolk of fresh eggs.

Egg Yolk TAC	Groups ^1^				SEM ^2^	*p*
	CTR	CI	CV	CVI		
Fresh yolk (% of ascorbic acid)	5.5 ^b^	11.8 ^a^	6.7 ^b^	12.3 ^a^	0.562	<0.001
Fresh yolk after oxidation challenge (% of ascorbic acid)	2.6 ^b^	8.3 ^a^	3.4 ^b^	9.1 ^a^	0.535	<0.001

^1^ CTR: not vaccinated and received drinking water without L + EO + VIT blend; CI: not vaccinated and received drinking water with L + EO + VIT blend; CV: vaccinated and received drinking water without L + EO + VIT blend; CVI: vaccinated and received drinking water supplemented by L + EO + VIT blend, (n = 4). ^2^ SEM: Standard Error of Mean. ^ab^ Different letters denote significant (*p* ≤ 0.05) differences between the dietary treatments.

**Table 8 vaccines-12-00147-t008:** Effect of dietary supplementation with the L + EO + VIT blend on total phenolic content after challenge with iron and ascorbic acid for 30 min to induce oxidation on egg yolk of fresh eggs.

Egg Yolk TP	Groups ^1^				SEM ^2^	*p*
	CTR	CI	CV	CVI		
Fresh yolk (μgGAE/g)	173.5 ^b^	233.2 ^a^	181.2 ^b^	245.1 ^a^	6.283	<0.001
Fresh yolk after oxidation challenge (μgGAE/g)	110.6 ^b^	208.3 ^a^	125.1 ^b^	201.2 ^a^	8.423	<0.001

^1^ CTR: Control diet; CI: Diet supplemented with L + EO + VIT blend; CV: Control diet plus bird vaccination; CVI: Diet supplemented with L + EO + VIT blend plus bird vaccination, (n = 4). ^2^ SEM: Standard Error of Mean. ^ab^ Different letters denote significant (*p* ≤ 0.05) differences between the dietary treatments.

**Table 9 vaccines-12-00147-t009:** Effect of dietary supplementation with the L + EO + VIT blend on protein carbonylation after challenge with iron and ascorbic acid for 30 min to induce oxidation on egg albumen of fresh eggs.

Egg Albumen Protein Carbonylation	Groups ^1^				SEM ^2^	*p*
	CTR	CI	CV	CVI		
Fresh yolk (nmol/mL)	13.7 ^a^	5.5 ^b^	15.7 ^a^	6.2 ^b^	0.832	<0.001
Fresh yolk after oxidation challenge (nomol/mL)	43.3 ^a^	21.9 ^b^	58.7 ^a^	22.3 ^b^	2.835	<0.001

^1^ CTR: Control diet; CI: Diet supplemented with L + EO + VIT blend; CV: Control diet plus bird vaccination; CVI: Diet supplemented with L + EO + VIT blend plus bird vaccination, (n = 4). ^2^ SEM: Standard Error of Mean. ^ab^ Different letters denote significant (*p* ≤ 0.05) differences between the dietary treatments.

**Table 10 vaccines-12-00147-t010:** Effect of vaccination and the L + EO + VIT blend on serum antibody titers in layers.

Antibody Titers %	Groups ^1^				SEM ^2^	*p*
	CTR	CI	CV	CVI		
**Before vaccination**						
NDV	4.300	4.296	4.320	4.293	0.010	0.839
IBV	4.170	4.100	4.186	4.146	0.030	0.822
**2-weeks post-vaccination**						
NDV	4.217 ^b^	4.325 ^b^	5.157 ^a^	5.317 ^a^	0.129	<0.001
IBV	4.087 ^b^	4.285 ^b^	5.357 ^a^	5.827 ^a^	0.189	<0.001
**4-weeks post-vaccination**						
NDV	4.270 ^b^	4.125 ^b^	4.440 ^b^	5.277 ^a^	0.115	<0.001
IBV	4.107 ^b^	4.346 ^b^	5.314 ^a^	5.455 ^a^	0.155	<0.001

^1^ CTR: Control diet; CI: Diet supplemented with L *+* EO + VIT blend; CV: Control diet plus bird vaccination; CVI: Diet supplemented with L + EO + VIT blend plus bird vaccination, (n = 16). ^2^ SEM: Standard Error of Mean. ^ab^ Different letters denote significant (*p* ≤ 0.05) differences between the dietary treatments.

**Table 11 vaccines-12-00147-t011:** Effect of dietary supplementation with the L + EO + VIT blend on the CBC and serum indices.

Haematological and Serum Parameters	Groups ^1^				SEM ^2^	*p*
	CTR	CI	CV	CVI		
TWBC (×10^3^/μL)	11.311 ^ab^	12.620 ^ab^	10.793 ^b^	13.671 ^a^	358.75	0.012
Heterophils (×10^3^/μL)	3.376	4.450	3.401	5.118	279.91	0.066
Lymphocytes (×10^3^/μL)	9.604	9.575	8.242	7.672	555.16	0.534
Monocytes (×10^3^/μL)	0.695	0.842	0.918	0.874	81.79	0.807
Eosinophils (×10^3^/μL)	0.154	0.218	0.190	0.194	20.44	0.759
Glucose (mg/dL)	273.75 ^a^	189.62 ^b^	279.75 ^a^	215.12 ^ab^	12.32	0.014
Albumin (g/dL)	2.187	2.300	2.362	2.075	0.082	0.639
Cholesterol (mg/dL)	148.37	136.75	113.62	130.87	10.90	0.743
Total Protein (g/dL)	6.475	6.262	5.537	5.762	0.193	0.294
ALP (IU/L)	255.75	295.87	196.75	456.50	50.17	0.307
ALT(IU/L)	10.375	7.500	9.875	8.000	0.578	0.225
AST (IU/L)	600.25	242.75	584.37	325.62	61.53	0.082
γ_GT (IU/L)	14.091	15.087	13.521	17.711	0.998	0.478
Cortisol (ng/mL)	0.978	0.943	0.910	0.846	0.033	0.565

^1^ CTR: Control diet; CI: Diet supplemented with L + EO + VIT blend; CV: Control diet plus bird vaccination; CVI: Diet supplemented with L + EO + VIT blend plus bird vaccination, (n = 8). ^2^ SEM: Standard Error of Mean. ^ab^ Different letters denote significant (*p* ≤ 0.05) differences between the dietary treatments.

**Table 12 vaccines-12-00147-t012:** Effect of dietary supplementation with the L + EO + VIT blend on serum oxidation status.

Serum Antioxidants	Groups ^1^				SEM ^2^	*p*
	CTR	CI	CV	CVI		
MDA (nmol/mL)	18.44 ^ab^	14.59 ^b^	22.53 ^a^	14.62 ^b^	1.073	0.017
CAT (U/mL)	1.540 ^x^	1.047 ^y^	1.354 ^x^	1.115 ^y^	0.071	0.058
SOD (U/mL)	1.373	1.221	1.397	1.373	0.036	0.311
GPx (nmol/mL)	0.235 ^a^	0.160 ^ab^	0.166 ^ab^	0.149 ^b^	0.012	0.034

^1^ CTR: Control diet; CI: Diet supplemented with L + EO + VIT blend; CV: Control diet plus bird vaccination; CVI: Diet supplemented with L + EO + VIT blend plus bird vaccination, (n = 8). ^2^ SEM: Standard Error of Mean. ^ab^ Different letters denote significant (*p* ≤ 0.05) differences between the dietary treatments. ^x,y^ Values in the same row with different superscripts tend to differ (0.05 < *p* ≤ 0.10).

**Table 13 vaccines-12-00147-t013:** Effect of dietary supplementation with the L + EO + VIT blend on liver and lung lesions.

Liver Evaluation	Groups ^1^				SEM ^2^	*p*
	CTR	CI	CV	CVI		
Hepatocellular necrosis	1.250	1.125	1.375	1.250	0.077	0.750
Hemorrhagic lesions	0.500	0.250	0.500	0.375	0.088	0.730
Microvascular necrosis	1.125	0.500	0.750	0.625	0.100	0.140
**Lung evaluation**						
Hemorrhagic lesions	0.500	0.375	0.750	0.500	0.100	0.626
Hyperplasia	0.500	0.250	0.875	0.375	0.100	0.140

^1^ CTR: Control diet; CI: Diet supplemented with L + EO + VIT blend; CV: Control diet plus bird vaccination; CVI: Diet supplemented with L + EO + VIT blend plus bird vaccination (n = 8). ^2^ SEM: Standard Error of Mean.

**Table 14 vaccines-12-00147-t014:** Effect of dietary supplementation with the L + EO + VIT blend on intestinal histomorphometry.

Histological Evaluation	Groups ^1^				SEM ^2^	*p*
	CTR	CI	CV	CVI		
Duodenum villus height	1365.8	1295.0	1297.5	1276.2	63.48	0.749
Duodenum crypt depth	135.1	138.5	135.5	123.33	6.59	0.869
Jejunum villus height	1736.8	1664.3	1679.4	1712.0	34.47	0.891
Jejunum crypt depth	261.72 ^a^	224.11 ^ab^	209.67 ^ab^	202.77 ^b^	8.00	0.034

^1^ CTR: Control diet; CI: Diet supplemented with L + EO + VIT blend; CV: Control diet plus bird vaccination; CVI: Diet supplemented with L + EO + VIT blend plus bird vaccination, (n = 8). ^2^ SEM: Standard Error of Mean. ^ab^ Different letters denote significant (*p* ≤ 0.05) differences between the dietary treatments.

**Table 15 vaccines-12-00147-t015:** Effect of dietary supplementation with the L + EO + VIT blend on the trachea histological lesion score.

Trachea Lesion Score	Groups ^1^				SEM ^2^	*p*
	CTR	CI	CV	CVI		
1-week post-vaccination	2.875 ^a^	2.750 ^a^	2.125 ^ab^	1.500 ^b^	0.170	0.009
2-weeks post-vaccination	2.500 ^a^	2.250 ^ab^	1.625 ^ab^	1.375 ^b^	0.141	0.009

^1^ CTR: Control diet; CI: Diet supplemented with L + EO + VIT blend; CV: Control diet plus bird vaccination; CVI: Diet supplemented with L + EO + VIT blend plus bird vaccination, (n = 8). ^2^ SEM: Standard Error of Mean. ^ab^ Different letters denote significant (*p* ≤ 0.05) differences between the dietary treatments.

## Data Availability

The data presented in this study are available on request from the corresponding author. The data are not publicly available due to privacy restrictions.

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
