# Peer review of "Effects of Dietary Supplementation of Essential Oils, Lysozyme, and Vitamins’ Blend on Layer Hen Performance, Viral Vaccinal Response, and Egg Quality Characteristics"

_vaccines, 2024, doi:10.3390/vaccines12020147_

Round 1

Reviewer 1 Report

Comments and Suggestions for Authors

1. There is great uncertainty in the composition of essential oils from different sources. The results of this study are based only on the specific composition and dosage of the essential oil used. Therefore, it is recommended to give a specific name, such as a commercial name for the essential oil used in this study to distinguish it from other essential oil products, and to not simply use "EO" to refer to the essential oil product used in this study.

2. The study did not measure the viscosity and dry matter weight of egg albumen.

3. The study uses ELISA to determine the titer of anti-NDV antibodies. Why not use the hemagglutination inhibitionHItest, which is widely used in veterinary clinics?

4. How was the dosage of L, EO and VIT determined for this study? Why this dosage? Is it based on the author's previous experiments? If so, please list the relevant literature in the references. If not, please explain the basis.

5. Chickens are small animals, in addition to the number of antibody titer test samples is 16, the detection of other indicators in this study, the samples collected by each experimental group, only from 8 chickens, which is too small. Of course, I agree with the number of samples in pathology tests.

Author Response

Dear Reviewer,

Thank you for your valuable feedback on our manuscript. We appreciate your suggestions and have made revisions accordingly. We believe these changes have significantly strengthened our work. We hope the updated version aligns with your expectations.

Reviewer 2 Report

Comments and Suggestions for Authors

This is an interesting study on the effect of dietary supplementation of natural products instead of antibiotics and vaccination against NDV on various parameters on poultry. The study is well designed and the results are well described and justified by the authors. Some minor comments to be addressed by the authors:

(a) The authors refer in the introduction the practice of use of prophylactic antibiotics in hens to reduce mortatlity and to improve productivity. The authors should mention that this practice also favors the expansion of drug resistant bacterial strains into the ecosystems which may affect directly or indirectly human populations. So there is an immediate interest to reduce the consumption of antibiotics and to this crucial point, since antibiotic consumption is directly associated with antimicrobial  resistance.

(b) The authors could add some information about the effect of NDV and IBV on flocks and some epidemiological data about those viruses since the readers of the journal may not be familiar with those animal viruses. 

Comments on the Quality of English Language

The paper is well written and minor editing is needed. 

Author Response

(The authors gave the same response as above.)

Reviewer 3 Report

Comments and Suggestions for Authors

This manuscript has been described on effects of dietary supplementation mixed with essential oils, lysozyme vitamins on lay hen. Although interesting, there are many unclear points to consider, as detailed below.

 1. A single experiment with four replicates is unlikely to provide good fidelity to the data. Therefore, it is necessary to confirm the accuracy of the experiment through repeated experiments.

 2. In this study, three dietary supplementations (essential oils, lysozyme, vitamins) were used. However, there are no any information on p-cymene, gamma-terpinene, linalool, and carvacrol in introduction section or materials and methods.

 3. Why choose these four molecules and this single dose for this study? Please add reasons based on papers, preliminary experiments, etc.

 4. In this study, the effect of dietary supplement containing essential oils, lysozyme and vitamins on laying hens was monitored, excluding the experimental group of each health supplement. Therefore, this design style does not accurately indicate which of the three molecules is operating. Accordingly, there is difficulty in using sentences like the following: “Various authors reported that EOs bioactive molecules function as free radical binders and inhibit lipid peroxidation, reducing the risk of oxidative stress and lipid oxidation [39]” in line 454.

 5. Each table needs sample numbers(n=???).

 6. In line 434, The sentences “According to our results, the performance of IBV and NDV vaccination in the flock of young laying hens had a positive effect on egg production three weeks after administration; although, for a short timeframe vaccination stress let to a minor decrease of the production rate. It has been also reported a positive effect of phytobiotics on egg production [32][33]” are not mated with Results. This manuscript focuses on the effects of dietary supplementations rather than the effects of IBV and NDV vaccination. Also, there is no relationship between your description and the references.

 7. According to Table 8, the statement in line 35: “In conclusion, water supplementation with L+EO+VIT enhanced humoral immune responses against IBV and NDV” is incorrect. Only one time point (week 4, NDV) showed increased antibody titers. There were no differences compared to CV at the other 3 time points. Therefore, 25% is effective, and 75% makes no difference, so the above expression needs to be modified.

 8. Since we are analyzing the effects of three dietary supplementations, we need to compare CTR and CI, and also compare CV and CVI for easier understanding. Comparing everything at once can be difficult to understand and cause confusion.

 9. Figure 1 is not cited in the text. Therefore, Figure 1 can be moved to the supplementary material after being cited in the main text.   

 10. In Table 13, CTR had the highest lesion score compared to the other three groups including CV. These data may indicate that the CTR group was infected with a pathogen during the study.

 11. Table 9 showed data of the blood analysis. In general, all animals have normal values (lowest and highest), so it is difficult to give significant statistical significance to values within this range.

Minor points:

 - In line 66, delete they.

-in line 77, According a to may be According to a.

-in line 246 and 247, means is mean.

- The scoring index in the legend of Figure 1 and the description at line 238 was inconsistent.

Author Response

(The authors gave the same response as above.)

Round 2

Reviewer 3 Report

Comments and Suggestions for Authors

The following comment 1 was not well responded. An explanation is needed that the experiment was not repeated. This does not mean measuring the same experiment repeatedly.

 1. A single experiment with four replicates is unlikely to provide good fidelity to the data. Therefore, it is necessary to confirm the accuracy of the experiment through repeated experiments.

Author response: We provided analysis for egg production that did not show significant differences among the experimental groups. Repeated measures would only marginally increase the results and conclusions as egg production was measured for a short period before and soon after the vaccination. All other measurements where only made on specific timepoints. In case of major differences were spotted; repeated measurements would be more appropriate in order to specify either the difference or the timepoint.

Author Response

We thank reviewer for his evaluation on our work. We believe that one study, for our work and for publishing resutls of one trial is adequate.

We agree with the comment of reviewer that a series of trials may provide more concrete results. We also know that EOs composition may show great variability among different oils types or even among the EOs of a specific plant due to geographical, cultivation and weather conditions. However, in our study we evaluated the effects of a commercial product consisted of EOs blend, Lysozyme and Vitamins manufactured to contain a stable composition being checked for each batch produced by the company, Patent Co. The company has conducted a large number of in vitro and in vivo trials before our application. For this reason, we want to add that the dosage and inclusion levels of L, EO and VIT have been based on previous findings by both  in vitro and in vivo experimentations of the PATENT Co company (4 of the co-authors are affiliated in the company).

We believe that we have a rather simple experimental design, and we provide conclusions based on our findings without exacerbating on further implications.